# Study of the Deep Processes of COVID-19 in Russia: Finding Ways to Identify Preventive Measures

**DOI:** 10.3390/ijerph192214714

**Published:** 2022-11-09

**Authors:** Alexander P. Anyutin, Tatiana M. Khodykina, Ekaterina I. Akimova, Elena V. Belova, Ekaterina A. Shashina, Denis V. Shcherbakov, Valentina V. Makarova, Nadezhda N. Zabroda, Anna A. Klimova, Nina A. Ermakova, Tatiana S. Isiutina-Fedotkova, Yury V. Zhernov, Roman V. Polibin, Oleg V. Mitrokhin

**Affiliations:** 1Institute of Radio Engineering and Electronics of the Russian Academy of Sciences, Fryazino Branch, 141190 Fryazino, Russia; 2Department of General Hygiene, F. Erismann Institute of Public Health, I.M. Sechenov First Moscow State Medical University (Sechenov University), 119435 Moscow, Russia; 3Department of Chemistry, Lomonosov Moscow State University, 119991 Moscow, Russia; 4Center for Medical Anthropology, N.N. Miklukho-Maclay Institute of Ethnology and Anthropology of the Russian Academy of Sciences, 119017 Moscow, Russia; 5Department of Epidemiology and Evidence-based Medicine, I.M. Sechenov First Moscow State Medical University (Sechenov University), 119991 Moscow, Russia

**Keywords:** COVID-19, new coronavirus infection, pandemic, preventive measures, epidemic, mobilization readiness, rapid response, wavelet analysis

## Abstract

The novel coronavirus disease 2019 (COVID-19) pandemic has had a huge impact on all areas of human life. Since the risk of biological threats will persist in the future, it is very important to ensure mobilization readiness for a prompt response to the possible emergence of epidemics of infectious diseases. Therefore, from both a theoretical and practical standpoint, it is currently necessary to conduct a thorough examination of the COVID-19 epidemic. The goal of this research is to investigate the underlying processes that led to the COVID-19 pandemic in Russia and to identify ways to improve preventive measures and ensure mobilization readiness for a quick response to potential COVID-19-like pandemics. This research will analyze the daily dynamics of the number of infection cases and the number of new lethal cases of COVID-19. We analyzed the daily number of new cases of COVID-19 infection N(d), the daily number of new lethal cases L(d), their percentage ratio L(d)/N(d) 100% in Russia for 2 years of the pandemic (from the beginning of the pandemic to 23 March 2022), the rate of increase and decrease of these indicators (dN(d)/dd and dL(d)/dd), as well as their spectra created on the basis of wavelet analysis. Wavelet analysis of the deep structure of the N(d) and L(d) wavelet spectra made it possible to identify the presence of internal cycles, the study of which makes it possible to predict the presence of days with the maximum number of infections and new deaths in a pandemic similar to COVID-19 and outline ways and methods for improving preventive measures and measures to ensure mobilization readiness for a rapid response to the potential emergence of pandemics similar to COVID-19.

## 1. Introduction

At the end of 2019, the Russian Federation, like the whole world, faced a global challenge associated with a new coronavirus disease 2019 (COVID-19) caused by the severe acute respiratory syndrome coronavirus 2 (SARS-CoV-2) that has spread regardless of state borders. The COVID-19 pandemic has had a huge impact on the social, economic, political, and cultural spheres of the country [1,2,3,4].

The first case of COVID-19 in Russia was fixed when a Moscow resident who had recently moved there from Italy was found to have COVID-19 on 2 March 2020 [5]. With the arrival of COVID-19, it has become clear that epidemics and pandemics will continue to pose a threat to society [5,6]. Because of this, it is very important to establish mobilization readiness for a quick response to the potential appearance of infectious disease epidemics and/or an increase in the risk to the public’s health and economic harm brought on by a pandemic like COVID-19 [6,7,8,9,10].

A thorough examination of the COVID-19 pandemic is currently a crucial endeavor from a theoretical and practical standpoint, as we noted earlier [11,12]. Many forecasting models are available, but some researchers note the limited accuracy of the models used for COVID-19 forecasts [13,14,15]. There are numerous strategies to address the biological, chemical, pharmaceutical, and other difficulties that arise in this condition. The dates, which are presented by the daily cases of infection, death, and recovery, are the most readily available material describing the dynamics of the pandemic. Colum bar’s graphs are typically used to display this type of data on a graph [16]. It is generally known that using the spectrum Fourier analysis approach is linked to the most popular method for analyzing time series of different origins [17,18]. However, spectral wavelet analysis, a substitute technique, has lately gained popularity for the study of time series [17,18,19,20]. For instance, Kondratiev’s economic series [21], El Niñio cycles in geophysics, changes in pulse dynamics and heart rhythmic activity in medicine, and hidden cycles in tomography [17,18,19,20] have all been studied using wavelet analysis.

The preferred use of Wavelet analysis is due to the fact that it has a significant advantage over Fourier analysis. This benefit results from the fact that Wavelet analysis can identify internal cycles in time series, whereas Fourier analysis cannot since the Fourier coefficients only capture information about the behavior of time series during the course of their existence.

The purpose of this work is to investigate the underlying processes in the development of the COVID-19 pandemic in Russia in order to find ways to improve preventive measures and ensure mobilization readiness for a prompt response to the possible emergence of pandemics similar to COVID-19.

## 2. Materials and Methods

The data for the wavelet analysis were obtained from an open database on the incidence of COVID-19 in Russia of the Federal Service for Surveillance on Consumer Rights Protection and Human Wellbeing in Russia (Rospotrebnadzor), the state sanitary service in Russia, which also transmits the same data to WHO [16].

We examined the daily number of new COVID-19 infections N(d), the daily number of new lethal cases L(d) reported in [16], their percentage ratio of 100 percent L(d)/N(d) in Russia for the two years of the pandemic (from the start of the pandemic to 23 March 2022), the rate of increase and decrease of these indicators (dN(d)/dd and dL(d)/dd), as well as their spectra created on the basis of wavelet analysis.

We also looked at the daily number of new lethal cases of COVID-19, which in this context can be viewed as an independent indication, since the number of new cases of COVID-19 infection may be connected to the number of tests conducted.

An essential element of the epidemiological process is the ratio of daily new cases of COVID-19 infection to daily new lethal cases because it can reveal the severity of the illness and, indirectly, the efficacy of preventive efforts.

Since the distribution of indicators is non-linear, the rate of increase and decline of the aforementioned indicators was estimated rather than the growth rate.

The dynamics of the number of new instances of infection, their rate of growth and decline, and their proportion were depicted in graphs. It was noted above that the dynamic for the daily number of infection and new lethal cases of a pandemic is usually depicted in the form of bar graphs [9]. For the convenience of analysis, we combined three graphs: the graph of the number of new cases of infection N(d)—Figure 1a, the graph of the number of new deaths L(d)—Figure 1b, and the graph of their percentage ratio 100% L(d)/N(d)—Figure 1c for the study period. Graphs of average daily cases of infection and new lethal cases from COVID-19 as a time function were obtained by the least squares method with using the sum of eight sinusoidal functions as a basis functions [17,18]. Graphs of the average velocity of new infections and new lethal cases from SARS-CoV-2 as a time function are the derivative of new infections dN(d)/dd and new lethal cases from COVID-19 dL(d)/dd.

The average percentage (%) ratio of daily new lethal cases to the number of daily new cases of COVID-19 infection was also obtained by the least squares method with using the sum of eight sines function as a basis functions.

We examined the spectrograms obtained on the basis of a wavelet analysis of time series, representing the daily dynamics of the number of daily new cases of infection N(d) and daily new lethal cases L(d) from COVID-19 in Russia, in order to study the internal structure of the complex process of development of the COVID-19 pandemic in that country. At the same time, for the convenience of obtaining spectrograms, the piecewise functions N(d) and L(d) were interpolated by cubic splines. As a result, we transformed the piecewise functions N(d) and L(d) into smooth functions. As wavelets, we used sixth order Daubechies wavelets (db6) [22,23]. The time of the wavelet spectra of pandemic development in days is plotted along the horizontal axis, and the Daubechies wavelets amplitudes (db6) at each of 16,384 levels are plotted along the vertical axis (at each level, the frequency of wavelet oscillations is constant and increases with increasing level number). Note that at a fixed level, the distances between the selected vertical lines of the same color correspond to the duration of inside cycles.

## 3. Results

### 3.1. Analysis of Changes in the Daily Number of Infections and the Dynamics of New Lethal Cases from COVID-19 in Russia over the Study Period

Figure 1′s graphs show information on the dynamics of new COVID-19 deaths and the daily number of infections in Russia for the two years under consideration (a, b, c).

These graphs (Figure 1a–c) depict wave-like fluctuations in the parameters under study. The progression of new lethal cases (Figure 1b) paralleled that of infections (Figure 1a), although lagged considerably behind the former. A maximum of new lethal cases followed a maximum of infections over the first 660 days. With a dramatic increase in infections starting on day 661, the number of new lethal cases dropped, as is particularly evident in Figure 1c.

These indicators’ charts show multiple important phases of growth and decrease, as well as numerous swings of varied sizes superimposed over larger ones.

In order to better understand the dynamics of the pandemic, we undertook a thorough investigation of changes in the daily number of infections and the dynamics of daily new death cases of COVID-19 in Russia over the research period.

### 3.2. Dynamic Analysis of Daily New Infections

Graphs of the function N(d), which indicate the time history of new instances of COVID-19 infection in Russia throughout the study period, are shown in Figure 2a,b. The bar graph, which was created using the information in [16], is indicated by the number 1. The continuous curve, shown by the number 2, was created by interpolating this data using the least squares approach with a basis function equal to the total of eight sinusoidal functions [17,18]. The continuous curve with the number 3 indicates how quickly the number of new infections is changing, or dN(d)/dd.

The graph of the functions of the number of new daily new infections N(d)—curve 1—contains both large- and small-scale variations (oscillations) with various periods, as can be seen from Figure 2a,b. Note that we presented the function N(d) in the form of two graphs, which are displayed in Figure 2a,b, respectively, due to a notable difference in the values of the function within 1 day < d < 660 day and within 661 days < d < 738 days. Curve 2 is the average time course of daily new infections (left scale), and curve 3 is a graph of the rate of change of this indicator (right scale).

If each section of the curve N(d) containing the rise and fall is conditionally referred to as a wave, then the process N(d) can be understood as a superposition of five local waves. There are three stages in each wave. An abrupt rise in the quantity of new infections characterizes the first stage. The second stage relates to the period of time that sees the highest numbers of new instances of infection. The third stage is a region of decrease that does not reach the initial level and fluctuates at relatively low numbers for a few days. Following this period, there is an increase in the number of new infections, which signals the arrival of the next wave. It can be assumed that during the decrease in the number of new infections from each previous wave, the number of new infections of each subsequent wave increases, which prevents the previous wave from reaching its minimum level. As a result, there is some wave “overlapping”.

We emphasize the existence of discernible small-scale changes in the graph of the daily number of new cases of infection at each stage, with varying durations and amplitudes.

Within 660 days of the observation’s start, Figure 2a depicts 4 waves, or periods of rise and fall in the number of new cases of infection. The highest number of new infection cases in the second wave increased as compared to the first wave at the same period. Although the “peak” of the third wave was a little lower than the “peak” of the second, the minimum numbers of new cases of infection between the third and fourth waves were higher than those of the first wave’s “peak” at that time. In comparison to the earlier investigated periods, the fourth wave exhibited faster growth and a higher “peak” in the number of daily new infections (Figure 2a, lines 2 and 3). Thus, we notice that with each wave there was a rise in the rate of increase in the number of daily new infections of COVID-19 and a decrease in the interval between waves.

It should be noted that the number of daily new infections itself turned out not to be the most revealing attribute, but rather the rate of change in the number of new infections.

Figure 2b shows the 5th wave on a separate graph, since the presentation of all five waves on one graph does not allow the small details of their changes to be distinguished due to the large difference in the number of daily new infections. Let us go into more depth about the latter. According to Figure 2a,b, the maximum number of new instances of infection at the “peak” of the fourth wave was greater than 40,000, and at the “peak” of the fifth wave, it was greater than 200,000, or five times more. The 5th wave of incidence began with a sharp jump, the rate of infection (an increase of 9000 people infected daily) exceeded even the most “high-speed” 4th wave by 15 times (an increase of 550 infected per day). Following a rapid ascent and a 2-day “peak” on the fifth wave, there was a rapid decline in the incidence that was faster than any preceding period. However, even at the “peak” of the first wave, the number of daily new infections on 23 March 2022, was much larger (26,826 and 11,230 infected, respectively).

It is important to note that the dynamics of new daily infections in the fifth wave, which was seen in January and February 2022, are very different from those in all earlier waves. (Figure 2a,b).

The difference Δ between the highest and minimum values of the rate of their change (Δ = max(dN(d)/dd)–min(dN(d)/dd)) separates all waves of new daily instances of infection significantly from one another.

We studied spectrograms based on wavelet analysis of time series, reflecting the daily dynamics of new daily instances of COVID-19 infection in Russia, in order to analyze the internal structure of the complicated process of development of the COVID-19 pandemic in that country (d).

It is clear from the data displayed on the spectrogram (Figure 3) that 4 distinct zones can be distinguished with different distances between the vertical lines. This indicates that there are 4 internal cycles within the spectrum, each with a different period: cycle 1 lasts for 3 days, cycle 2 for 8 days, cycle 3 for 20–22 days, and cycle 4 for 90–100 days. The presence of cycles and their interference can lead to complex fluctuations in new cases of daily COVID-19 infection and allows you to predict the presence of days with a maximum number of this indicator.

### 3.3. Study of the Dynamics of Daily Number of New Lethal Cases

A graph of the function L(d), which represents the temporal course of daily number of new lethal cases in Russia from the start of the pandemic to 23 March 2022, is shown in Figure 4.

Graphs of the function L(d), showing the progression over time of the daily number of new COVID-19-related fatal cases in Russia during the study period: Line 1 shows the actual time course of new COVID-19-related fatal cases; Line 2 shows the average time course of new fatal cases; and Line 3 shows a graph showing the rate of change of the average time course of new COVID-19-related deaths.

Please take note that there are five waves in the graphs depicting the daily number of new deaths from COVID-19 in Russia. These graphs can be compared to graphs of the number of new infections to see that, over time, the waves of lethal cases lag behind the waves of infections somewhat. According to Figure 1a,b, the number of infections and the maximum values of the first four waves of new deadly cases both show a constant upward trend. The maximum number of new deaths from COVID-19 and the rate of their increase was noted in the 4th wave of the pandemic. Although there is a noticeable delay in the number of new lethal cases relative to the number of infections in the 5th wave, there is also a noticeably lower maximum value than in the 4th wave. Keep in mind that the dynamics of the rate of increase in the daily number of new deaths from COVID-19 are similarly similar: the “peaks” of the largest increase occur between the first and fourth waves and dramatically decline on the fifth wave.

For a more in-depth investigation, we created and looked into the spectrogram of the daily number of new fatalities from COVID-19, which was derived on the basis of wavelet analysis (Figure 5).

The analysis of the obtained spectrogram also identified four internal cycles with distinctive periods: cycle 1–3 days, cycle 2–8 days, cycle 3–20–22 days, and cycle 4–90–100 days. These cycles coincide with the internal cycles of the dynamics of the daily number of new cases of COVID-19 infection N(d).

### 3.4. Analysis of a Time Series Illustrating the Daily Dynamics of the Ratio of Daily New COVID-19 Deaths to Daily New Infection Cases

The ratio of daily new COVID-19 deaths to daily new infections, 100 percent L(d)/N, is one of the key factors influencing the pandemic’s progress (d). This is due to the fact that such a characteristic may indicate the severity of the disease and indirectly the effectiveness of preventive measures.

Figure 6 illustrates the graphs of the change in the percentage of daily new deaths from COVID-19 to the number of daily new cases of infection in Russia over the study period 100% L(d)/N(d), as well as the velocity of change of this ratio d [100% L(d)/N(d)] /d*d*.

Figure 6 demonstrates how the ratio of COVID-19′s daily new deaths to its daily new infection rate, 100 percent L(d)/N(d), similarly fluctuated in waves. Note how the first four waves’ changes in this ratio closely reflect the graphs of new instances of infection N(d) and new cases of mortality L(d).

However, the differences in their behavior are as follows: compared to the second wave N(d), the second wave 100 percent L(d)/N(d) has a substantial delay, and the maximum of waves 2–4 of the ratio are quite similar, the fifth the wave is significantly different in the presence of a sharp decline and a minimum of the ratio of 100% L(d)/N(d) against the background of maxima of new cases of daily infection N(d) and daily new deaths L(d) of the fifth wave of the COVID-19 pandemic; likewise, the amplitude of small-scale oscillations in the first two waves of the ratio 100 percent L(d)/N(d) is substantially bigger than that of the waves that follow in the third and fourth waves.

Additionally, we observe that the graph of the rate of change of this ratio (see curve 3 in Figure 6) has nearly the same maxima, setting it apart from dependences for daily new lethal cases L and daily new infection cases N that are identica L(d).

## 4. Discussion

We conditionally referred to these periods of rise and fall as waves in accordance with the data of the Federal Service for Surveillance on Consumer Rights Protection and Human Wellbeing in Russia (Rospotrebnadzor) [24] by analyzing changes in the time course of the daily number of new cases of COVID-19 infection N(d), the time course of the daily number of new deaths from COVID-19 L(d), and their ratio over 2 years. Five waves can simultaneously be identified in Russia from the start of the pandemic until 23 March 2022. As a result of continuous particular and general preventive actions, in our perspective, this confirms the heterogeneity and shift in the prevalent genovariant of COVID-19. This is consistent with the data of Rospotrebnadzor [24], making it urgent to find ways and strategies for a more rapid response.

From mid-April to June 2020, the incidence of COVID-19, the so-called “1st wave of COVID-19,” increased, according to Rospotrebnadzor. Incidence of COVID-19 increased again from October to December 2020, coinciding with the “2nd wave of COVID-19.” The “3rd wave of COVID-19” was described as an increase in COVID-19 incidence from May to June 2021. This increase may have been caused by the emergence of a new mutant strain of the SARS-CoV-2 Delta (VOC21APR-02, B.1.617.2) in the Russian Federation, which took the place of the alpha strain of the SARS-CoV-2 VOC-202012/01, B.1.1.7 or 20B/501Y. V1 in the current structure of circulating strains [24].

As a result, while the number of new cases of infection from each previous wave decreases, the number of new cases of infection from each subsequent wave increases, preventing the previous wave from reaching its minimum level, leading to a kind of “superposition” of waves. This supports our conclusion that the circulating SARS-CoV-2 strains change in each subsequent wave [25,26].

The Russian Federation put in place a number of organizational measures to stop the spread of SARS-CoV-2 infection in the critical period from the start of the threat of COVID-19 spreading until May 2020. This complex consisted of three stages.

Rospotrebnadzor initially implemented a number of preventative and anti-epidemic actions. The Russian Ministry of Health and health officials prepared the medical network for a massive influx of patients during the second stage. The third stage saw the implementation of measures by the state to self-isolate citizens.

It is important to take note of the implementation of general disease prevention and control measures among health professionals, as well as the use of non-specific preventative measures (such as masks, gloves, isolation, and disinfection measures, the transfer of enterprises and institutions to a remote form of work, etc.). These activities have reduced the number of daily new cases of COVID-19. The positive effect of these general prevention measures can be seen in a decline in the ratio of the number of daily new deaths from COVID-19 to the number of daily new cases of infection (characterizing the severity of the clinical manifestation disease) observed from 200 to 300 days of the COVID-19 pandemic, even though later, when restrictive measures were lifted, an increase in new cases and the formation of a second wave were observed. However, as the number of daily new fatalities and COVID-19 infection cases decreased from day 300 to day 430, this ratio started to rise. This could be because the Delta genovariant, which has a more severe clinical course, has started to spread in the Russian Federation starting March 2021. The Delta genovariant dominated the Russian Federation by the end of August 2021. The Delta Plus genovariant then appeared. These genovariants could cause the third and fourth waves of the pandemic because of their high infectivity. These waves’ durations, however, were shorter than that of the first and second waves. In other words, both the number of new infections per day and the number of COVID-19 deaths per day decreased more quickly. This characteristic can be attributed to the development of herd immunity as a result of population vaccination. The dynamics of the ratio of daily new COVID-19 deaths to daily new infections from day 430 to day 650 of the pandemic in Russia also support this. An essential element of the epidemic process is the ratio of daily new cases of COVID-19 infection to daily new fatalities because it can indicate the severity of the illness and, indirectly, the efficacy of preventive efforts [27].

It is interesting to note that during the first four pandemic waves in Russia, the dynamics of changes in the ratio of daily deaths to daily infections from COVID-19 were comparable to the dynamics of the time course of daily new infections N(d) and daily deaths from COVID-19. The fifth wave of the COVID-19 pandemic, however, is considerably different since it has a steep decrease and a minimum of the ratio of 100 percent L(d)/N(d) against a background of highs in new cases of daily infection N(d) and daily new fatalities L(d). The change of the dominant genovariant to Omicron, which is characterized by a higher infectivity and a milder clinical course as well as specific and nonspecific preventative interventions, can likely be attributed to this characteristic of the fifth wave [28,29].

In general, there was a trend toward a rapid decline in the daily number of new infections and deaths from COVID-19 when examining the five waves of the COVID-19 pandemic in Russia. We believe that this might be seen as a positive result of a set of preventive measures, especially in the formation of herd immunity [30,31]. This is further supported by a decline in the ratio of daily new deaths to COVID-19 infection cases in waves three through four, particularly in wave five.

Research on the dynamics of COVID-19 in Germany also confirms the importance of vaccination: unvaccinated individuals were involved in 8–9 out of 10 new infections [32]. The study in Italy and Israel also found a positive impact of the vaccination campaign on the dynamics of the COVID-19 pandemic [33,34]. The importance of creating sufficient immunity to influence the dynamics of COVID-19 is also shown by a study in the United States [35].

According to the study, the average time course of the daily number of new COVID-19 infection cases’ rate of change provides more useful information than the daily number of new cases’ dynamics. This indicator can be suggested while preparing preventive actions in the context of a quick reaction to changes in the epidemic scenario.

We created and examined spectrograms of the daily number of new COVID-19 infection cases and deaths, which were produced using wavelet analysis, for a more thorough examination. As a result, four internal cycles with different periods were identified: cycle 1–3 days, cycle 2–8 days, cycle 3–20–22 days, and cycle 4–90–100 days. Cycles’ existence and interference with one another might cause complex oscillations in the researched indicators. The results of these wavelet analysis can be utilized to predict when days with a large volume of daily infection new cases and COVID-19 mortality will occur.

It should be noted that there are many other analyzes of the dynamics of the spread of COVID-19. Talmoudi K. et al. study to assess the dynamics of transmission of COVID-19, maximum likelihood estimation (ML) and Akaike information criterion (AIC) were performed for serial interval distributions, namely normal, lognormal, Weibull, and gamma [36]. The calculation of serial intervals (SI) included a significant number of non-positive values, so the authors had to fit the distributions to both positive values (truncated) and biased data, in which 12 delays were added to each observation. Thus, it should be taken into account that there are no grounds for estimates and forecasts based on the data obtained by the authors, with the exclusion of negative values.

Another study examined exponential growth and estimated epidemic doubling time and base reproductive number based on information about the demographics, exposure history, and timing of laboratory-confirmed COVID-19 cases reported up to January 22, 2020. Case characteristics were described, and major epidemiological distributions of time delays were assessed. These initial findings were drawn from a “linear list” of 10 cases and are somewhat inaccurate; for further research, the authors noted the need to provide more information about this distribution [37].

Nishiura H. et al. study collected onset dates for primary cases (infectious) and secondary cases (infected) from published research articles and case investigation reports [38]. The authors used a Bayesian approach with a doubly censored likelihood interval to obtain consistent interval estimates. The Widely Applicable Information Criterion (WAIC) was used to compare distributions, and the model with the lowest WAIC was chosen as the best fit model for each set of scores with and without right truncation.

To analyze the dynamics of infection, the authors of another study used the distribution of serial intervals of COVID-19 since they are an important starting point for determining the basic reproduction number (R0) and the amount of intervention required to control the epidemic [39].

In another study, the authors evaluated the serial interval of the COVID-19 [40]. To fit three different distributions, gamma, Weibull, and lognormal, which govern the calculation of serial intervals (SI) of COVID-19. The distribution was chosen according to the Akaike information criterion, adjusted for small sample size (AICc). The lognormal distribution was found to perform slightly better than the other two distributions in terms of AICc. Assuming a lognormally distributed model, they estimated the mean SI. 

Using a parameterized susceptible-infected-infected-recovered model, the authors of another study modeled the dynamics of the spread of the COVID-19 outbreak and the impact of various control measures on it [41]. The SEIR algorithm was applied, which analyzes the flows of people between four states: susceptible (S), exposed (E), contagious (I) and recovered (R). Given that only a few studies have been conducted on R0, the authors tried to take into account the dynamic changes in R values under different epidemic development scenarios.

The advantage of our wavelet analysis compared to previously described works is that we were able to identify internal cycles of COVID-19 in time series, while other analyzes have this limitation. This wavelet analysis can serve as the foundation for the creation of a mathematical model of the COVID-19 pandemic in order to provide preventive measures.

With regard to obtaining information on prevention and control measures based on wavelet analysis, it was shown that the internal cycles of the COVID-19 pandemic are periodic. This means that if we know any point in the dynamics of the development of the COVID-19 pandemic, then we can expect it to recur after a period characteristic of this internal wave. In turn, this means that it is possible to make an approximate forecast of the days with the maximum number of cases of daily COVID-19 or the maximum daily number of new deaths from COVID-19.

In our study of the internal cycles of the COVID-19 pandemic in Russia, wavelet analysis showed that there are 4 internal cycles with different periods in the spectrum: cycle 1–3 days, cycle 2–8 days, cycle 3–20–22 days, and cycle 4–90–100 days. Note that 3-day and 8-day cycles may be associated with different incubation periods for different SARS-CoV-2 strains. A cycle of 90–100 days can indicate the frequency of the days with the maximum number of new cases of daily COVID-19 infection or the maximum number of new cases of daily deaths from COVID-19. For example, at the beginning of the next rise in the number of new cases of daily COVID-19 infection (or new cases of daily deaths from COVID-19), knowing the duration of the internal cycles of the pandemic, it is possible to roughly predict the dynamics of the development of the pandemic and the time of the onset of days with the maximum number of cases of daily infection (or death), which makes it possible to prepare preventive measures in advance, for example, introducing a mask regime, self-isolation regime/lockdown, medical observation, or quarantine.

Therefore, based on wavelet analysis, it is urgent to develop methods and strategies for a more prompt response and mobilization readiness for a prompt response to the potential emergence of epidemics of infectious diseases similar to COVID-19. In order to develop ways to enhance models for the implementation of preventative and anti-epidemic actions for comparable illnesses, it is necessary to further investigate the causes of the features that have been identified. This is of both theoretical and practical interest.

## 5. Conclusions

We provisionally referred to five significant periods of growth and decline in these indicators as waves after analyzing the dynamics of the daily number of new infections and new fatalities from COVID-19 in Russia. The “superposition” of the following wave on the prior one, as seen in this instance, may be caused by a change in the SARS-CoV-2 strain. While the number of illnesses from the previous strain declines, the number of infections from the next strain rises.

We also found that with each new wave of the pandemic, there is a tendency for the number of new infections and fatalities from COVID-19 to decline more quickly. This might point to an effective COVID-19 pandemic response in Russia.

Wavelet analysis of the deep structure of the COVID-19 pandemic in Russia discovered four internal cycles with different periods: 3 days, 8 days, 20–22 days, and 90–100 days. The existence of internal cycles makes it possible to predict the days with the maximum number of infections and new deaths in a pandemic similar to COVID-19. Further research is needed to build a pandemic development model that is similar to COVID-19 and to develop effective preventive strategies based on the characteristics of the COVID-19 pandemic in Russia.

## Figures and Tables

**Figure 1 ijerph-19-14714-f001:**
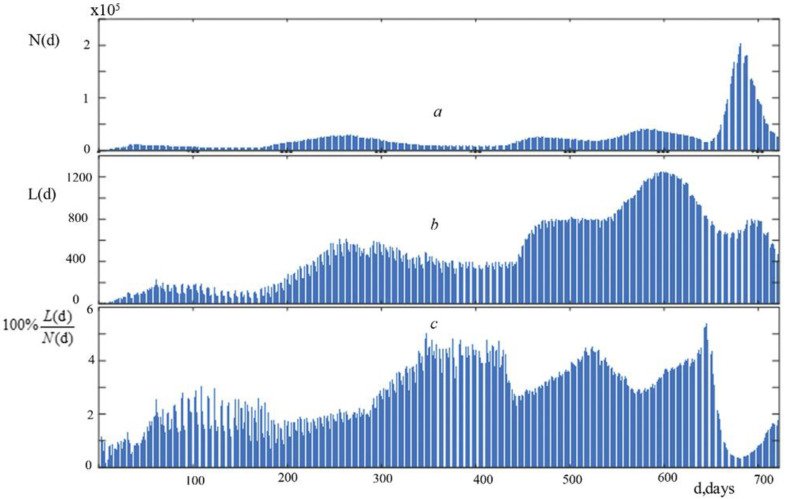
(**a**)—time variance for the daily new number of infections N(d)in Russia; (**b**)—time variance for the daily new number of death L(d) in Russia; (**c**)—time variance for the relative death 100% L(d)/N(d) in Russia.

**Figure 2 ijerph-19-14714-f002:**
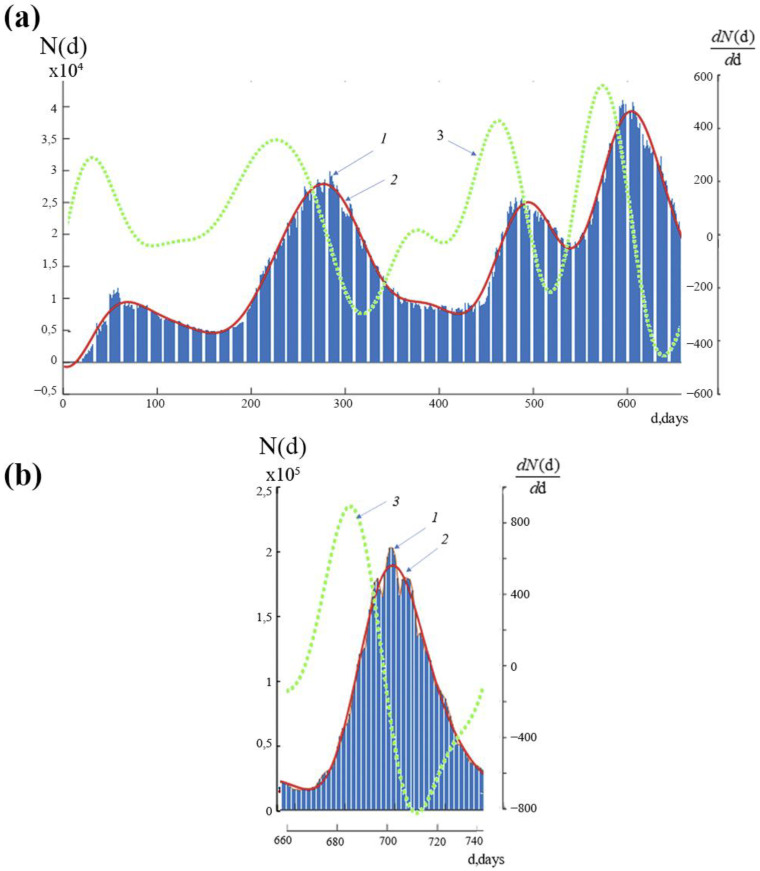
(**a**)—time variance for the daily new number of infections N(d) in Russia within period: 1 day < d < 660 day; 1—the real dates of new cases of COVID-19; 2—the average dates of new cases of COVID-19; 3—the velocity of the average dates of new cases of COVID-19; (**b**)—time variance for the daily new number of infections N(d) in Russia within period: 661 days < d < 738 days; 1—the real dates of new cases of COVID-19; 2—the average dates of new cases of COVID-19; 3—the velocity of the average dates of new cases of COVID-19.

**Figure 3 ijerph-19-14714-f003:**
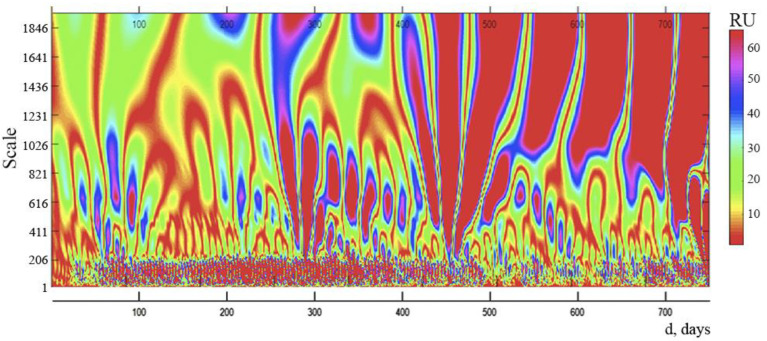
Wavelet spectrum of a continuous function N(d), representing the time course of new cases of daily COVID-19 infection in Russia during the study period. The color scale determines the amplitude of the spectrum in relative units (RU).

**Figure 4 ijerph-19-14714-f004:**
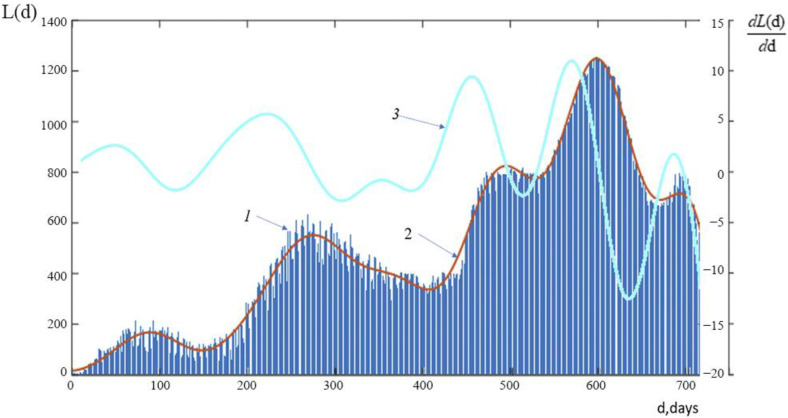
Time variance for the daily number of new lethal cases in Russia. 1—the real dates of new cases of death from COVID-19; 2—the average dates of new cases of death from COVID-19; 3—the velocity of the average dates of new cases of death from COVID-19.

**Figure 5 ijerph-19-14714-f005:**
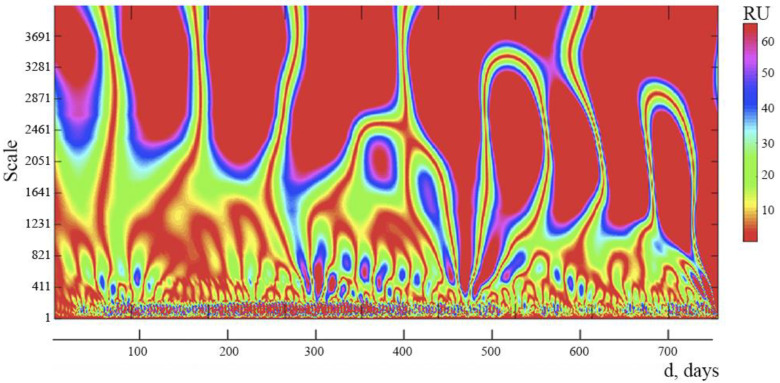
The wavelet spectrum of the daily number of new deaths from COVID-19 in Russia. The color scale determines the amplitude of the spectrum in relative units (RU).

**Figure 6 ijerph-19-14714-f006:**
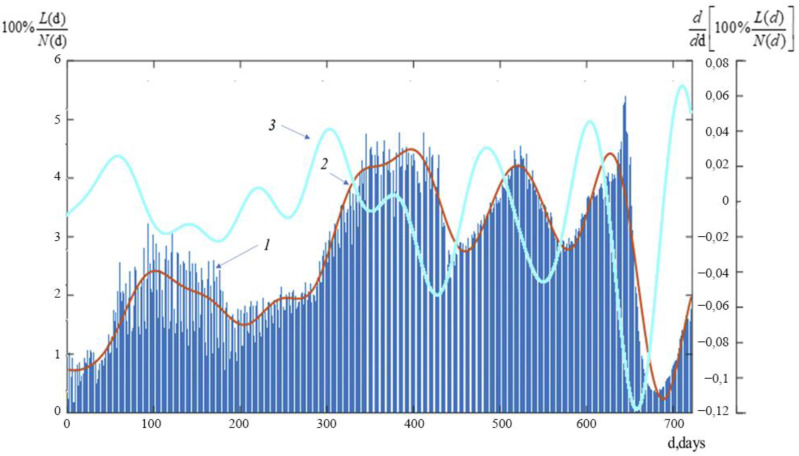
The relative daily deaths in Russia. 1—the real dates of relative death from COVID-19 in Russia in %; 2—the average dates of relative death from COVID-19 in Russia in %; 3—the velocity of the average dates of relative death from COVID-19 in Russia in %.

## Data Availability

The data presented in this study are available on request from the corresponding author.

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
