# Peer review of "Study of the Deep Processes of COVID-19 in Russia: Finding Ways to Identify Preventive Measures"

_ijerph, 2022, doi:10.3390/ijerph192214714_

Round 1

Reviewer 1 Report (Previous Reviewer 2)

Thank you for the revised version of the manuscript, which includes references to alternative models to explore and describe the epidemiological pattern of the pandemic. The wavelet analysis sheds light on the internal cycles of the Covid epidemics, but the article is short in exploring how this analysis can inform prevention and control measures, as proposed in the conclusion.

Author Response

Dear Reviewer,

Thank you for allowing us to resubmit our manuscript "Study of the deep processes of COVID-19 in Russia: finding ways to identify preventive measures" for publication in the IJERPH MDPI. We appreciate the time and effort you dedicated to providing feedback on our manuscript and are grateful for the insightful comments about prevention and control measures. We have incorporated minor corrections and submitted the manuscript. Please see below for our response to the comment.

Authors' Responses

Thank you for your comment!

We have added the necessary information about prevention and control measures based on our wavelet analysis. Please find the added text in the Discussion chapter highlighted in yellow (Lines 474–493) and also here below:

‘With regard to obtaining information on prevention and control measures based on wavelet analysis, it was shown that the internal cycles of the COVID-19 pandemic are periodic. This means that if we know any point in the dynamics of the development of the COVID-19 pandemic, then we can expect it to recur after a period characteristic of this internal wave. In turn, this means that it is possible to make an approximate forecast of the days with the maximum number of cases of daily COVID-19 or the maximum daily number of new deaths from COVID-19.

In our study of the internal cycles of the COVID-19 pandemic in Russia, wavelet analysis showed that there are 4 internal cycles with different periods in the spectrum: cycle 1 – 3 days, cycle 2 – 8 days, cycle 3 – 20–22 days, and cycle 4 – 90–100 days. Note that 3-day and 8-day cycles may be associated with different incubation periods for different coronavirus strains. A cycle of 90–100 days can indicate the frequency of the days with the maximum number of new cases of daily COVID-19 infection or the maximum number of new cases of daily COVID-19 deaths. For example, at the beginning of the next rise in the number of new cases of daily COVID-19 infection (or new cases of daily COVID-19 deaths), knowing the duration of the internal cycles of the pandemic, it is possible to roughly predict the dynamics of the development of the pandemic and the time of the onset of days with the maximum number of cases of daily infection (or death), which makes it possible to prepare preventive measures in advance, for example, introducing a mask regime, self-isolation regime/lockdown, medical observation, or quarantine.’

Sincerely yours,

Yury Zhernov

Professor of the Department of General Hygiene,

I.M. Sechenov First Moscow State Medical University (Sechenov University)

This manuscript is a resubmission of an earlier submission. The following is a list of the peer review reports and author responses from that submission.

Round 1

Reviewer 1 Report

The authors presented a well-written manuscript that requires improvement. The first limitation is the lack of information about obtaining data for the presented analysis. Which unit in Russia has reported Covid-19 results and are the results reliable? Information on consents to use data should be included in the manuscript.

Another indication for the Authors is the improvement of the Discussion section. It should begin with a summary of the results of your own research and then compare the results with other researchers around the world. Conclusions should not be included in points. This is a bad methodological practice. It should be clearly described.

Author Response

Thank you very much for your review of the manuscript and comments!

We have added changes to the manuscript, highlighted in yellow. In the 'Materials and Methods' chapter, a source for obtaining data was added - this is an open public database of The Federal Service for Surveillance on Consumer Rights Protection and Human Wellbeing (Rospotrebnadzor). The same data are submitted from Russia to WHO.

As for the conclusion, another reviewer asks to leave it point by point. We ask you to agree to this option. At the same time, we reworked the points, reducing their number and describing them in more detail.

Reviewer 2 Report

The use of wavelet analysis brings a new perspective on time series dynamics of incident events and case-fatality rates from COVID 19. However, since this analytical model is largely unknown in the field of traditional medical epidemiological surveillance and reporting, it is very important to highlight the additional practical contribution of this approach. Although the authors attribute event waves and internal cycles to shifts in the prevalence of coronavirus strains and to the possible impact of control measures, it is unclear from the study results and discussion to what extent this information can guide future preventive measures and preparedness the authors claim. The authors need to explore how the wavelet analysis and the resulting spectrograms can be used regularly with other time series methods to better characterize the dynamics of epidemics and how to exploit their potential added value. We appreciate the summary conclusions presented in bullet point form, so the readers can understand the interpretation of the results.

Author Response

Thank you very much for reviewing our manuscript and for your valuable advice!
We made the necessary changes to the manuscript and highlighted them in yellow.

Round 2

Reviewer 1 Report

The authors added information about data obtained from open database on the incidence of COVID-19 in Russia of the Federal Service for Surveillance on Consumer Rights Protection and Human Wellbeing in Russia. They didn't change the Conclusions form. Unfortunately, the discussion does not follow the form adopted for scientific publications around the world. The authors did not undertake any discourse about the situation in any other European country. It is necessary to add a comparison for the publication to gain scientific value. I am asking for a scientific discussion on the topic undertaken in other European countries, the USA or the situation of China. The literature includes 17 items, including two by non-Russian authors, which is unacceptable in an international scientific journal and may constitute bias. Please cite other authors. It is a bad practice to not have a letter to the reviewers. Please send Author's reply for reviews.